Computed tomography analysis of guinea pig bone: architecture, bone thickness and dimensions throughout development

Witkowska Agata 1
Alibhai Aziza 1
Hughes Chloe 1
Price Jennifer 1
Klisch Karl 1
Sturrock Craig J. 2
Rutland Catrin S. 1 Catrin.rutland@nottingham.ac.uk
1 School of Veterinary Medicine and Science, University of Nottingham , Sutton Bonington, Leicestershire , UK
2 School of Biosciences, University of Nottingham , Sutton Bonington, Leicestershire , UK
Hutchinson John
Electronic publication date: 2014 Oct 2
Publication date: 2014
Volume: 2
Electronic Location ID: e615
Received 2014 Sep 4; Accepted 2014 Sep 18
Copyright: © 2014 Witkowska et al.
Copyright year: 2014
Copyright holder: Witkowska et al.
License: This is an open access article distributed under the terms of the Creative Commons Attribution License, which permits unrestricted use, distribution, reproduction and adaptation in any medium and for any purpose provided that it is properly attributed. For attribution, the original author(s), title, publication source (PeerJ) and either DOI or URL of the article must be cited.
License URL: https://creativecommons.org/licenses/by/4.0/

Keywords: Micro computed tomography, Guinea pig, Anatomy, Development, Bone, Supratrochlear foramen, Nutrient foramen

Funding: The School of Veterinary Medicine, University of Nottingham BBSRC summer studentship Funding was provided to CSR by The School of Veterinary Medicine, University of Nottingham. A BBSRC summer studentship funding was also provided to CSR to give JP research training. The funders had no role in study design, data collection and analysis, decision to publish, or preparation of the manuscript.

==============================
The domestic guinea pig, Cavia aperea f. porcellus, belongs to the Caviidae family of rodents. It is an important species as a pet, a source of food and in medical research. Adult weight is achieved at 8–12 months and life expectancy is ∼5–6 years. Our aim was to map bone local thickness, structure and dimensions across developmental stages in the normal animal. Guinea pigs (n = 23) that had died of natural causes were collected and the bones manually extracted and cleaned. Institutional ethical permission was given under the UK Home Office guidelines and the Veterinary Surgeons Act. X-ray Micro Computed Tomography (microCT) was undertaken on the left and right scapula, humerus and femur from each animal to ascertain bone local thickness. Images were also used to undertake manual and automated bone measurements, volumes and surface areas, identify and describe nutrient, supratrochlear and supracondylar foramina. Statistical analysis between groups was carried out using ANOVA with post-hoc testing. Our data mapped a number of dimensions, and mean and maximum bone thickness of the scapula, humerus and femur in guinea pigs aged 0–1 month, 1–3 months, 3–6 months, 6 months–1 year and 1–4 years. Bone dimensions, growth rates and local bone thicknesses differed between ages and between the scapula, humerus and femur. The microCT and imaging software technology showed very distinct differences between the relative local bone thickness across the structure of the bones. Only one bone showed a singular nutrient foramen, every other bone had between 2 and 5, and every nutrient canal ran in an oblique direction. In contrast to other species, a supratrochlear foramen was observed in every humerus whereas the supracondylar foramen was always absent. Our data showed the bone local thickness, bone structure and measurements of guinea pig bones from birth to 4 years old. Importantly it showed that bone development continued after 1 year, the point at which most guinea pigs have reached full weight. This study is the first to show the high abundance (100% in this study) of the supratrochlear foramen within the guinea pig humerus and the complete absence of a supracondylar foramen, which is different to many other species and may also affect potential fracture points and frequencies. Understanding bone morphology and growth is essential in not only understanding the requirements of the healthy guinea pig, but also necessary in order to investigate disease states.

Introduction

The domestic guinea pig (Cavia aperea f. porcellus) belongs to the Caviidae family of rodents (Burnie, 2008) that includes subfamilies covering species closely related to guinea pigs such as the Patagonian cavy, also known as the Mara (Dolichotis patagonum) or the world’s largest rodent, the capybara (Hydrochaerus hydrochaeris) (Burnie, 2008). The guinea pig has been kept as an important source of food, and is still eaten in many parts of South America, Asia and Africa (Meredith & Redrobe, 2010; Morales, 1995; NRC, 1991). Peru alone has around 20 million guinea pigs, providing around 17 thousand tonnes of meat per annum, just 4,000 tonnes less than their sheep meat production (NRC, 1991). The meat can fetch higher prices than pork or beef on small mountain farms in such regions as Ecuador (NRC, 1991). It has a relatively high protein and low fat content in comparison to other alternatives such as chicken, which makes it a good nutritional enrichment for many lower socio-economic families (Numbela & Valencia, 2003).

Guinea pigs are commonly kept as pets, as well as used extensively in medical research playing a pivotal role in epidemiological study and pharmaceutical development (Terril & Clemons, 1997). The population of guinea pigs used in research has declined from 2.5 million in the 1960s to just over 200,000 in 2010 (Gad, 2013; USDA, 2011), but the popularity of the guinea pig as a pet has soared. The number of guinea pigs kept as pets in the UK has consistently been estimated at between 0.5 and 1 million since 2009, with the guinea pig presently listed as the UK’s 8th most popular pet and highest ranking rodent (PFMA, 2013).

Adults can achieve their full weight of up to 1,800 grams but more usually between 900–1,200 grams for boars and 700–1,100 grams for sows (Behrend, 2008; Meredith & Redrobe, 2010) at 8–12 months, a higher weight than that of wild guinea pigs (Hubrecht & Kirkwood, 2010), however these values vary greatly across the literature and can only be used as a rough guide. As a hystricomorph, the guinea pig sow has a characteristically long gestation period of 59 to 72 days, approximately double that of the rabbit (Meredith & Redrobe, 2010). Litter size varies from 2 to 6, with an average of three or four pups, each weighing between 40 and 120 grams (Terril & Clemons, 1997). The precocial pups are born mobile, fully-furred, with their eyes open and teeth present, and are therefore able to consume solid food within a few hours, although still suckle for two to three weeks (Hubrecht & Kirkwood, 2010; Meredith & Redrobe, 2010). Although there is wide variability across the literature, it is believed that puberty is reached at around six weeks in sows and between 9 and 10 weeks in boars (Harkness et al., 2010). Life expectancy is generally considered to be between around 5–6 years (Mitchell & Tully, 2009), although life expectancies of up to eight years are reported (Gad, 2013).

Amongst their many other functions such as mineral storage or bone marrow production, bones are the main levers in mammalian bodies (Bilezikian, Raisz & Martin, 2008) enabling the animal to move. Bones have an additional secondary role in homeostasis as a store of calcium and phosphorus. The bone is first formed as cartilage and becomes mineralised during the final stages of pregnancy, a process that continues through to puberty and beyond, suggesting that the skeleton of an older animal is able to withstand higher forces than that of a neonate (Sjaastad, Hove & Sand, 2010). The bone consistency changes constantly throughout life as it undergoes remodelling in response to physical and metabolic factors which can affect density as well as volume (Frost, 1997; Sjaastad, Hove & Sand, 2010).

Despite the significance of the guinea pig within the food and pet industry there has been little research into the normal bone structure and density/thickness. One method that enables analysis of bone characteristics is Computed Tomography (CT). Since its invention in the 1970s (Hounsfield, 1973), CT is now widespread in clinical imaging, permitting non-destructive and non-invasive quantitative measurements of the body of both humans and animals. The imaging technique is based on the attenuation of X-rays as they penetrate the material of interest at a known number of angular positions. Subsequently, tomographic reconstruction algorithms are used to generate a three dimensional spatial map of X-ray attenuation of a material which can then be analysed in detail using computer software (Ritman, 2011). Within the last 10 years, microCT systems have become more common for non-clinical applications and offer higher spatial resolution, detail detectability and contrast compared to conventional CT (Metscher, 2009). The technique has been successfully applied to the investigation of bone development and anatomy in a range of small animals including the guinea pig, mouse and rat (Bialek et al., 2014; Tao et al., 2014; Uzun, Curthoys & Jones, 2007; Willett et al., 2012).

Long bones are composed of an outer layer of dense compact bone and an inner meshwork of trabecular bone, which is particularly abundant in the epiphyses, and bone marrow (Zoetis et al., 2003). In comparison, flat bones, such as the scapula, consist of two thick layers of compact bone with a layer of trabecular bone in between. The blade of the scapula is said to undergo intramembranous ossification, in which the bone develops from a fibrous membrane, whereas some of the outer parts undergo endochondrial ossification whereas long bones such as the humerus and femur form via endochondral ossification, where cartilage is replaced by bone (Ross & Pawlina, 2011; Scheuer, Black & Cunningham, 2000). Regardless of its method of formation, bone growth in length happens via cartilaginous growth plates, which fuse in later life, and in diameter by periosteal apposition, enabling the bone to withstand increasing loads (Ross & Pawlina, 2011). Density can also vary throughout the bone, with the shaft of long bones, for example, having a greater density than their extremities (Stiner, 2004).

Our study investigated the bone growth and localised thickness of the guinea pig scapula, humerus and femur. The rate of growth and the localised thickness are not known for these bones, despite the frequency at which guinea pigs present in veterinary clinics with broken/fractured limb bones. The scapula, along with the humerus, forms the shoulder joint. Similarly to the cat, but unlike other domestic species, the scapula has a small clavicle attaching it to the manubrium sterni. The scapula is a triangular shape and divided by a spine that runs over its lateral surface, into a supraspinous and infraspinous fossa (Dyce, Sack & Wensing, 2010). The distal part of the scapula ends in the glenoid cavity, which serves as a surface for articulation with the head of the humerus, forming the shoulder joint (Dyce, Sack & Wensing, 2010). The humerus is comprised of the head, contributing to the shoulder joint, and the articular condyle, part of the elbow joint (Dyce, Sack & Wensing, 2010). Other notable features include the greater and lesser tubercle, the medial and lateral epicondyles and the olecranon fossa (König & Liebich, 2014). The femur articulates with the pelvis via its head forming the hip joint, and caudally with the tibia via its condyles to form the stifle joint (Dyce, Sack & Wensing, 2010). It is the strongest long bone of the skeleton and comprises the proximal part of the hind limb with notable features including the greater and lesser trochanter and the trochanteric fossa (König & Liebich, 2014). This study investigates the attributes and growth of three key limb bones, the scapula, humerus and femur. Clinicians and health care advisors alike frequently highlight the importance of not dropping the guinea pig due to the number of limb fractures and breaks that they observe in clinic (Richardson, 2003). This study aims to increase the information known about the limb bones, whilst showing how and when development is occurring, in addition to providing localised bone thickness information.

The aims of this study were to utilise microCT technology to measure guinea pig bone dimensions and map bone local thickness in neonates through to adulthood. Although present literature states that adult weight is achieved at 8–12 months (Hubrecht & Kirkwood, 2010), bone growth and local thickness has not been elucidated despite its importance to guinea pig husbandry and clinical care. It is important to highlight that this paper investigates female guinea pigs for a number of reasons. Differences in bone development have been attributed to sex and neuter status (May, Bennett & Downham, 1991; Perry, Fordham & Arthurs, 2014; Root, Johnston & Olson, 1997) but the additional strain of lactation and pregnancy in the female may further decrease calcium content of bones (Horwits & Smith, 1990) and may therefore increase the potential of bone damage in this sex. In the pet and meat industry a greater number of female guinea pigs are present. For example an increased number of females are observed in the meat industry, with around 1 male to every 12 females generally accepted as the norm (Koeslag, 1989; Nuwanyakpa et al., 1997). Guinea pigs are herd animals in the wild and live in family units with a dominate male, but males that are strangers will frequently fight, therefore they are more difficult to house together as pets (Donnelly, 2010), resulting in pet owners being more likely to choose a male/female or female/female pair, or a group of females. It should also be highlighted that males may not show the same growth rates or bone thickness, however the full weight of 900–1,200 g for males as opposed to 700–1,100 g for the females, is still achieved at 8–12 months (Behrend, 2008; Hubrecht & Kirkwood, 2010; Meredith & Redrobe, 2010). Three methods of bone dimension measurement were utilised—a traditional manual calliper method, a manual measurement of microCT images and an automated microCT analysis. A further aim was to assess the location and number of nutrient foramina and the course of the nutrient canal in every bone, and report on the abundance of both the supratrochlear foramen and the supracondylar foramen in the humerus.

Materials and Methods

Sample collection

Naturally deceased, entire female guinea pigs with known medical and husbandry backgrounds were collected under ethical permissions obtained from The University of Nottingham in accordance with the British Home Office laws and the Veterinary Surgeons Act. All animals were fed on standard, commercially available guinea pig food ad. lib. Bones were extracted using manual dissection and grouped according to age—0–1 month (<1 m; n = 5), 1–3 months (<3 m; n = 4), 3–6 months (<6 m; n = 4), 6 months–1 year (<1 yr; n = 5) and 1–4 years (<4 yr; n = 5). The right and left scapula, humerus and femur from each guinea pig were analysed.

MicroCT and bone analysis

Three types of bone measurement were carried out (1) manual bone measurements as described in Fig. 1 using World Precision (UK) digital callipers calibrated to three decimal places. (2) Bone measurements as described in Fig. 1 using micro CT images and software. (3) Automated microCT measurements to find the maximal height, depth, width, surface area and volume. Measurements 1 and 2 were compared to ensure no discrepancy between manual and automated methods.

Figure 1 Gross anatomical photographs indicating measurements calculated.

(A) Scapula; 1, length of the bone; 2, length of the spine; 3, width from cranial to caudal angle; 4, width of the shoulder joint. (B) Humerus; 1, length from the head to the elbow; 2, width from the head to the greater tubercle; 3, width of the elbow joint and (C) Femur; 1, overall length from the head to the medial condyle.

Prior to scanning, individual bones were carefully wrapped in thin sheets of X-ray transparent polyethylene packing foam and placed in 40 mm diameter × 50 mm height plastic specimen jars. Depending on the size of the bones, each jar could accommodate up to 12 individual bones. Each jar was scanned using a GE Phoenix Nanotom S, X-ray microCT system (GE Sensing and Inspection Technologies GmbH, Wunstorf, Germany). The scan consisted of the collection of 1,200 angular projection images in ‘Fast’ mode at an electron acceleration energy of 110 kV and 160 µ A current. The resulting spatial resolution of the scan to fit the entire pot in the field of view was 24.24 µm. Scans were performed in approximately 30 min. Following tomographic reconstruction using Datos rec v1.5 (GE Sensing and Inspection Technologies GmbH, Wunstorf, Germany), individual bones were virtually extracted (segmented) from the 3D volumetric data based on their higher X-ray attenuation from the low density packing materials using a combination of object calibration and region growing tools in VGStudioMax V2.2. software (Volume Graphics GmbH, Heidelberg, Germany). The bone measurements (see Fig. 1), volume and surface area of the extracted volumes was measured automatically using the isosurface calibration values in the software. Bone maximal length and width was manually measured using the calliper tool in VGStudioMax XY image stack data for each bone was exported in 8 bit grayscale tiff image format. Bone local thickness was measured using the BoneJ (Doube et al., 2010) plugin for the open source image quantification and analysis software ImageJ 1.44 (Schneider, Rasband & Eliceiri, 2012). Compact bone local thickness heat map images were visualised in VGStudioMax. Bone dimensions, volumes and densities were analysed using ANOVA using SPSS (V16; f value 0.778, alpha 0.05, power 80%) statistical software, P < 0.05 was considered as a significant difference.

Nutrient foramina and canals and the humeral supratrochlear and supracondylar foramina were assessed using both sequential scan X-rays and 3D reconstructions. Nutrient foramina were classified as a cavity that fully breached the entire bone wall, the location of each foramen was recoded alongside the course of the nutrient canal through the bone. Each humerus was also assessed for the presence of both a supratrochlear foramen and a supracondylar foramen.

Results and Discussion

Bone development

A previous study compared CT scanning measurements to standard scientific callipers or assessing the skull base and the craniomaxillofacial dimensions in five humans (Citardi et al., 2001). CT measurements were found to be more accurate and had a better representation of bone anatomy, however, there was little significant difference in results, with P values ranging from 0.06 to 1.0 (Citardi et al., 2001). In order to ensure that a similar situation was observed in the smaller guinea pig bone, both microCT and manual measurements were carried out on the scapula, humerus and femur. Our data showed non-significant (P > 0.05) variations ranging from 96%–110% for each measurement, with 60.5% of the data falling within 5% range of the mean. The larger differences, although non-significant, were observed on the femur (smallest bone), specifically the length from the head to the medial condyle of the femur (also one of the smallest measurements). Therefore CT data was used to compare bone growth data (Fig. 2). In addition to data being presented in Figs. 2–9, raw data is supplied in Tables S1–S5.

Figure 2 Manual measurements aided by micro-CT.

Measurements of guinea pig bones aged 0–1 month (<1 m), 1–3 months (<3 m), 3–6 months (<6 m), 6 months–1 year (<1 yr) and 1–4 years (<4 yr). (A) Scapula, (B) humerus and (C) femur with associated ANOVA with post-hoc P value tables. NS, not significant. Mean ± standard error of the mean error bars.

Figure 3 Micro-CT scapula measurements.

Measurements of guinea pig scapula aged 0–1 month (<1 m), 1–3 months (<3 m), 3–6 months (<6 m), 6 months–1 year (<1 yr) and 1–4 years (<4 yr). (A) bone width, depth and length, (B) surface area and (C) volume with associated ANOVA with post-hoc P value tables. NS, not significant. Mean ± standard error of the mean error bars. SA, surface area.

Figure 4 Micro-CT humerus measurements.

Measurements of guinea pig humerus aged 0–1 month (<1 m), 1–3 months (<3 m), 3–6 months (<6 m), 6 months–1 year (<1 yr) and 1–4 years (<4 yr). (A) bone width, depth and length, (B) surface area and (C) volume with associated ANOVA with post-hoc P value tables. NS, not significant. Mean ± standard error of the mean error bars. SA, surface area.

Figure 5 Micro-CT femur measurements.

Measurements of guinea pig femur aged 0–1 month (<1 m), 1–3 months (<3 m), 3–6 months (<6 m), 6 months–1 year (<1 yr) and 1–4 years (<4 yr). (A) bone width, depth and length, (B) surface area and (C) volume with associated ANOVA with post-hoc P value tables. NS, not significant. Mean ± standard error of the mean error bars. SA, surface area.

Figure 6 Micro-CT bone local thickness heat mapping reconstructions.

Representative heat mapping reconstructions of guinea pig bone localised thickness aged 0–1 month (<1 m), 3–6 months (<6 m) and 1–4 years (<4 yr). Scapula, anterior (A) and posterior (B); humerus, anterior (C) and posterior (D); femur, anterior (E) and posterior (F). Scale bars for each age represent 8 mm.

Figure 7 Micro-CT bone local thickness measurements.

Measurements of guinea pig bone local thickness aged 0–1 month (<1 m), 1–3 months (<3 m), 3–6 months (<6 m), 6 months–1 year (<1 yr) and 1–4 years (<4 yr). (A) scapula, (B) humerus and (C) femur with associated ANOVA with post-hoc P value tables. NS, not significant. Mean ± standard error of the mean error bars.

Figure 8 Bone growth and localised thickness throughout development.

Percentage bone growth (A) and local thickness (B) of 0–1 month (<1 m), 1–3 months (<3 m), 3–6 months (<6 m), 6 months–1 year (<1 yr) old guinea pigs in comparison to 1–4 year olds.

Figure 9 Gross anatomical features of the humerus.

Humerus anterior (top row) and posterior (lower row) views showing examples of the supratrochlear foramen (as indicated by closed arrowhead) at 0–1 month (<1 m), 1–3 months (<3 m), 3–6 months (<6 m), 6 months–1 year (<1 yr) and 1 year–4 years (<4 yr). Examples of nutrient foramina in a 4 year old humerus are indicated (open arrowhead). Scale bars represent 8 mm.

Significant differences (P < 0.05 to P < 0.0001) in scapula measurements were observed between all ages, for all four measurements (as described in Fig. 1, values given in Fig. 2A and Table S1) except between 1 year and 4 year old guinea pigs in the width from cranial to caudal angle and width of the elbow joint (Fig. 2A, Tables S1 and S2, measurement 2, P > 0.05). The maximal width and length were significantly increased at 4 years old in comparison to 1 year old (Fig. 3A, P < 0.05) as were the scapula volume and surface area (Figs. 3B and 3C, P < 0.004 and P < 0.0001 respectively).

Significant differences in humeri measurements (as described in Fig. 1) were observed between all ages in all measurements (measurements shown in Fig. 2B and Tables S1 and S3, P < 0.008 to P < 0.0001) with the exception of width between the head and greater tubercle between 1 year and 4 years and the width of the elbow joint between 3 and 6 months. Overall however, the maximal width, depth, length, volume and surface areas did not significantly increase between 1 and 4 years old (Fig. 4, P > 0.05), therefore differing growth rates were observed in comparison to the scapula.

The femur measurements showed significant differences in measurements described in Fig. 1 between the different age groups (Figs. 2C, 5 and Tables S1, S4, P < 0.018 to P < 0.0001). Although the length from the head to the medial condyle and maximal width increased from 1 year to 4 years old, the maximal length and depth did not (Figs. 2C and 5A). In contrast to the humerus though, the volume and surface area of the femur were significantly increased from 1 year to 4 years (Figs. 5B, 5C and Table S4, P < 0.004 and P < 0.0001 respectively).

Mean scapula local thickness (Figs. 6, 7 and Table S5) significantly increased until 6 months old (P < 0.003), but appeared to stabilise thereafter before a slight decrease was observed at 4 years old, whereas the maximum scapula local thickness in younger guinea pigs was significantly lower in animals aged 6 months in comparison to 1 year/4 year old bones (Fig. 7A, P < 0.005 to P < 0.0001). Humerus local thickness, both mean and maximum, also significantly increased from 1 month to 4 years (P < 0.0001 and P < 0.002 respectively), but only the mean significantly increased in the latter stages from 6 months to 4 years (Fig. 7B, P < 0.023). Femur local thickness mean and maximum significantly increased until 1 year (P < 0.002) and 6 months (P < 0.033) respectively but no differences were observed beyond these ages (Fig. 7C).

The differential compact bone local thickness can be observed in detail across the developing scapula, humerus and femur in Fig. 6 and Table S5. It was interesting to note that higher growth rates were observed in the scapular (flat bone) between years 1 and 4 in comparison to the long bones (Fig. 8A, see also Figs. 2–5), however the bone local thickness was more likely to increase between these time points in the long bones in comparison with the scapula (Fig. 8B, see also Fig. 7). Despite these overall increases, it was noted that the older bones had a far greater variation in localised bone thickness, resulting in greater extremes of thickness being observed in comparison to younger bones (Figs. 6–8 and Table S5).

Nutrient, supratrochlear and supracondylar foramina

The nutrient foramen and canal are the result of the invading nutrient artery during fetal development (Payton, 1934). It was noted in our study that every bone had between 2 and 5 nutrient foramina; no bones contained a singular foramen. Within the scapula, the primary foramen is commonly located in most species at the lateral aspect of the infraspinous fossa, whilst secondary foramina (if present) can be located on either the infrasinous fossa or the subscapular fossae (Scheuer, Black & Cunningham, 2000). This was also observed in our study, but secondary foramina were also located on the scapula spine and acromion. Within the guinea pig long bones (femur and humerus), the primary nutrient foramina were located in the proximal or distal third of the diaphysis. Further foramina (up to four more per bone) were located: above the trochlear foramen and on the greater tubercle (humerus), and on the head and the shaft of the head (femur). This positioning is similar to most animals where within long bones, the primary nutrient foramen is often located in either the proximal or distal third of the diaphysis, with the canal running obliquely into the medulla (Payton, 1934). Previous research in the pig found that in the humerus, the foramen is located in the distal third of the bone and the canal directed proximally, while in the femur the foramen is located in the proximal third and the canal runs distally (Payton, 1934). In relation to the multiple foramina (Fig. 9) observed in the guinea pig bones, it has been reported that each bone generally has a single nutrient foramen, however studies on the canine femur found that only 6.2% of cases contained a single foramen, while more frequently, two or more foramina existed in a single bone (Ahn, 2013). Our serial X-ray data showed that every nutrient canal, regardless of bone, position or age, travelled in an oblique direction in the female guinea pig. The oblique direction observed concurs with, and supports, previous studies in both the guinea pig and most other species (De Bruyn, Breen & Thomas, 1970; Payton, 1934).

The lower end of the humerus has two large fossae, the olecranon fossa and the coronoid fossa, separated by a thin bony plate that rarely bears an opening known as supratrochlear foramen (STF). A STF was observed in all of the female guinea pig humeri (n = 46), left and right regardless of age (Fig. 9). The SFT is formed when the olecranon fossa at the caudal aspect of the distal humerus is so deep that it meets the radial fossa on the bone’s cranial aspect. The SFT is closed by a membrane of connective tissue and no major blood vessels of nerves pass through the foramen. The presence of the SFT within mammals is very variable and is linked to the range of mobility in the elbow joint. Most hystricognath rodents have a STF, which is likely to facilitate full extension of the elbow joint in terrestrial locomotion, while in arboreal species the olecranon fossa is more often shallow and a SFT is not formed (Candela & Picasso, 2008). In humans there is some variability in the presence of an SFT. A study of the humerus in North Indians found that the presence of the STF varied within the sampled population from not being present at all, to present bilaterally or only in one femur (Mahajan, 2011). 26% of the humeri had an STF, with a higher frequency in females and in the left humerus (Mahajan, 2011). It has been hypothesised that the presence of an STF may produce stress, altering the fracture lines and possibly increasing supracondylar fracture rates, even in low-energy trauma (Sahajpal & Pichora, 2006). This could be of clinical concern in guinea pigs as 58% of all human paediatric elbow fractures are in the supracondylar area of the humerus (Houshian, Mehdi & Larsen, 2001).

It was also noted that the supracondylar foramen was absent in all 46 guinea pig humeri. The supracondylar or entepicondylar foramen is a foramen proximal to the medial epicondyle of the humerus. The median nerve and brachial artery pass through this foramen. The supracondylar foramen is an ancestral structure in mammals and has been lost independently in several mammalian clades during mammalian evolution (Polly, 2007). The foramen is absent in the guinea pig, as well as in other hystricognath rodents (Candela & Picasso, 2008).

Conclusions

Understanding normal bone growth and development in the guinea pig is essential, especially in relation to movement and homeostasis as a store of calcium and phosphorus. Guinea pigs suffer from a number of bone disorders including metabolic bone disease – conditions that develop following prolonged calcium or vitamin D deficiency, or an improper ratio of calcium to phosphorus in the diet (Terril & Clemons, 1997). Imbalances have resulted in reports of reduced growth rate, pathological changes of the animal’s skeleton and osteodystrophia fibrosa resulting from nutritional secondary hyperparathyroidism (Rapsch Dahinden et al., 2009; Schwarz et al., 2001). Bone disorders can also manifest from hypovitaminosis C, a dietary insufficiency of vitamin C for just 2 to 3 weeks can result in lameness or pain due to intra-articular haemorrhage, anorexia, weight loss and general unthriftiness, progressing to death if untreated (Gad, 2013; Richardson, 2003; Terril & Clemons, 1997) and if combined with vitamin E deficiency the time can drop to just a few days (Hill et al., 2003). Disorders such as osteoporosis, common in this species (Bendele, White & Hulman, 1989), accelerate bone composition and modelling changes and decrease the bone’s density significantly, however this can also be attributed to normal age-related changes (Bilezikian, Raisz & Martin, 2008). In order to compare and understand disease states, normal bone growth, development and morphology must be categorised and our study has added to the data available on normal bones throughout development.

Differences in bone development have also been attributed to sex and neuter status (May, Bennett & Downham, 1991; Perry, Fordham & Arthurs, 2014; Root, Johnston & Olson, 1997), therefore it is important to highlight that the male guinea pig rates of growth and bone development characteristics may differ in relation to our study, as may neutered females. In addition to this, further variation may be seen within the female as lactation and pregnancy are known to alter calcium content of bones (Horwits & Smith, 1990).

Over the years a number of techniques have been used to investigate bone and unravel the complex biology of disorders, for example clinical CT, magnetic resonance and histology. MicroCT holds some benefits such as higher resolution or speed of sample processing in comparison with other techniques but with the caveat that it can only be used in non-living specimens (due to significantly increased X-ray radiation dose required for microCT in relation to clinical CT scanning).

Our study is the first to show the growth rates and local bone thicknesses for the scapula, humerus and femur in the developing guinea pig. Despite sexual maturity occurring at around 6–10 weeks, and full animal weight being achieved at 8–12 months, the scapula, humerus and femur continue to grow, and the local bone thicknesses alter beyond 1 year. This paper is also the first to show the absence of the supracondylar foramen and the unusual presence of the supratrochlear foramen in the humerus of this species, which may in turn affect fracture rates and locations within the humerus. Understanding guinea pig bone development and anatomy can help inform clinical and husbandry practice, especially in relation to bone thickness and fractures.

Supplemental Information

Supplemental Tables Manual measurements aided by micro-CT. Table S2. Micro-CT scapula measurements. Table S3. Micro-CT humerus measurements. Table S4. Micro-CT femur measurements. Table S5. Micro-CT bone local thickness measurements.

Click here for additional data file.

Supplemental Information 2 Ethical approval

Click here for additional data file.

The authors would like to thank the owners of the guinea pigs for donating their animals in order to further the understanding of this species. We would also like to thank the School of Veterinary Medicine and Science technical team for providing sample storage and space for the study.

Additional Information and Declarations

Competing Interests

Author Contributions

Animal Ethics

The authors declare there are no competing interests.

Agata Witkowska conceived and designed the experiments, performed the experiments, analyzed the data, wrote the paper, prepared figures and/or tables, reviewed drafts of the paper.

Aziza Alibhai performed the experiments, wrote the paper, reviewed drafts of the paper.

Chloe Hughes analyzed the data, wrote the paper, prepared figures and/or tables, reviewed drafts of the paper.

Jennifer Price and Karl Klisch analyzed the data, wrote the paper, reviewed drafts of the paper.

Craig J. Sturrock conceived and designed the experiments, performed the experiments, analyzed the data, contributed reagents/materials/analysis tools, wrote the paper, reviewed drafts of the paper.

Catrin S. Rutland conceived and designed the experiments, performed the experiments, analyzed the data, contributed reagents/materials/analysis tools, wrote the paper, prepared figures and/or tables, reviewed drafts of the paper, grant writing, ethical review submission, project management and funding.

The following information was supplied relating to ethical approvals (i.e., approving body and any reference numbers):

Naturally deceased guinea pigs were used however ethical permission was applied for. An approval letter was sent from the University of Nottingham committee to the PI confirming that the use of naturally deceased pet animals could be collected and utilized and ethical permission was given for the entire project.

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
