# Peer review of "Computed tomography analysis of guinea pig bone: architecture, bone thickness and dimensions throughout development"

_PeerJ, doi:10.7717/peerj.615_

## Round 0.1 · original submission · Minor Revisions

The reviewer has some final requests for improvements that I concur with- while abstracts vary at PeerJ it is more normal to have a cohesive single paragraph than split into dry intro-methods-results-discussion sections there.

That the analysis includes only females and the reason for this does need to be justified up front, ideally at the end of the Introduction. I have no problem with it though, and I think the reviewer agrees as long as it is justified more explicitly.

As for their point 2, I agree it could be a bit clearer why the bones must be measured- I agree these data are valuable but 1-2 sentences more in the Intro toward the end could better convince readers of the value of these data, and then a stronger reminder of what the findings mean toward the end of the Discussion/Conclusion, which could include any emphasis of novel findings (but it seemed to me that this is an area in which there is little if any study of this kind so it may largely be novel!).

If these simple revisions are done, I will accept the paper. Thanks for hanging in there- we want to ensure that adequate quality controls are enforced as well as fairness to you and to past/future authors and reviewers.

Reviewer 1 ·

Basic reporting

1. Please provide a brief abstract instead of provide the abstract for each section of the paper.

Experimental design

1. What's the rational include all female guinea pigs as the samples? Is there any difference between male and female bones?

2. What's the significance of measuring bones of guinea pigs?

3. What's the novel findings compared to previous studies?

Validity of the findings

No Comments

---

## Round 0.2 · accepted · Accept

These are adequate revisions- I have checked them all and the reviewer's requests have been satisfied. Hence I can now accept the paper- well done!